# Case based rubella surveillance in Abia State, South East Nigeria, 2007–2011

Chukwuemeka Anthony Umeh[1] and Stella Chioma Onyi[2]

[1] Department of International Health, Boston University School of Public Health, MA, USA
[2] New York Institute of Technology, College of Osteopathic Medicine, Old Westbury, NY, USA

## ABSTRACT

**Introduction.** Rubella infection has the potential of causing severe fetal birth defects collectively called congenital rubella syndrome (CRS) if the mother is infected early in pregnancy. However, little is known about rubella and CRS epidemiology in Nigeria and rubella vaccines are still not part of routine childhood immunization in Nigeria.

**Methods.** Analysis of confirmed cases of rubella in Abia State, Nigeria from 2007 to 2011 detected through Abia State Integrated Disease Surveillance and Response system.

**Results.** Of the 757 febrile rash cases, 81(10.7%) tested positive for rubella immunoglobulin M (IgM). New rubella infection decreased from 6.81/1,000,000 population in 2007 to 2.28/1,000,000 in 2009 and increased to 6.34/1,000,000 in 2011. The relative risk of rubella was 1.5 (CI [0.98–2.28]) times as high in females compared to males and 1.6 times (CI [0.90–2.91]) as high in rural areas compared to urban areas. Eighty six percent of rubella infections occurred in children less than 15 years with a high proportion of cases occurring between 5 and 14 years.

**Conclusion.** Rubella infection in Abia State, Nigeria is predominantly in those who are younger than 15 years old. It is also more prevalent in females and in those living in rural areas of the state. Unfortunately, there is no surveillance of CRS in Nigeria and so the public health impact of rubella infection in the state is not known. Efforts should be made to expand the rubella surveillance in Nigeria to incorporate surveillance for CRS.

## INTRODUCTION

Rubella is an acute contagious viral infection that usually affects children and young adults. It is a mild self-limiting illness that presents with fever, maculopapular rash, malaise and mild conjunctivitis. However, 20–50% of all rubella infections occur without a rash, or are subclinical. Rubella is transmitted by airborne droplets when infected people sneeze or cough (*WHO, 2013a*; *WHO, 2011*).

Rubella is of public health importance because of its teratogenic effect on the fetus if the mother is infected in early pregnancy or just before conception. A woman who is infected just before conception or in early pregnancy has a 90% chance of having her fetus infected with rubella. This can lead to miscarriage, stillbirth or severe birth defects which are collectively known as congenital rubella syndrome (CRS) (*WHO, 2013a*; *WHO, 2011*).

Corresponding author
Stella Chioma Onyi,
umehs@hotmail.com

CRS can present with major eye and heart defects and hearing impairments. Children with CRS are also at risk of developing serious developmental disabilities and are prone to other lifelong disabilities such as autism, diabetes mellitus and thyroid dysfunction (*WHO, 2011*).

Globally, an estimated 110,000 children are born annually with CRS with most of the CRS occurring in South East Asia, Africa, and Western Pacific regions—regions where few countries have introduced rubella containing vaccines as part of the national vaccination schedule. Due to massive rubella vaccination, the incidence of rubella and CRS has been greatly reduced or has practically been eliminated in most developed countries and some developing countries (*WHO, 2011*; *WHO, 2013b*).

Rubella vaccines are live attenuated vaccines and a single dose of the vaccine confers long-lasting immunity in more than 95% of the vaccine recipients. There is also naturally induced immunity after rubella infection. The presence of immunoglobulin G (IgG) antibodies in the serum is indicative of rubella immunity, although there are variations from country to country in the level of IgG that is considered protective, rubella IgG antibodies $\geq$ 10 IU/ml is generally considered protective (*WHO, 2011*; *WHO, 2013b*).

There has been a renewed effort by World Health Organization (WHO) to eliminate measles and rubella in most regions of the world. This led to the launching of a new Global Measles and Rubella Strategic Plan by the Measles & Rubella Initiative in April 2012. The plan aims to eliminate measles and rubella in at least five WHO regions by 2020. Part of the ways to achieve this include: vaccination coverage with two doses of measles- and rubella-containing vaccines; effective disease surveillance, and building up public confidence and demand for immunization (*WHO, 2013a*).

The rubella vaccine is not part of the routine immunization schedule in Abia State, Nigeria (*Abia State Government, 2013a*). There has also been lack of attention on rubella in Abia State, Nigeria. Abia State is in the south-eastern part of Nigeria and covers a land area of 5,243.7 square kilometers. The population of Abia State by the 2006 population census was 2,833,999 and with an annual growth rate of 2.7%, the estimated population in 2012 is 3,368,430. Abia State has 17 local government areas (*Abia State Government, 2013b*; *UNFPA, 2012*). About 70% of the population lives in rural areas. The state has a high proportion of young population with children aged 0–14 years accounting for 36.8% of the population. There is also a high age dependency ratio of 66.5%. Over 59% of the population is estimated to live below the poverty line of one US dollar a day (*UNFPA, 2012*).

This article gives a descriptive analysis of the rubella cases detected through the Abia State Integrated Disease Surveillance and Response between 2007 and 2011.

## METHOD

Cases of rubella in Abia State, South East Nigeria from January 1, 2007 to December 31, 2011 detected through the Abia State Integrated Disease Surveillance and Response were analyzed. The data was collected from 130 disease surveillance focal sites in Abia State. These sites are evenly distributed around the state and are involved in both active and passive surveillance of febrile rash in communities and health facilities. The focal persons

**Table 1 Number of specimens that tested positive for Rubella Immunoglobulin (Ig) M Antibodies out of all the specimens tested, Abia State, Nigeria, 2007–2011.**

| Year of onset | Total number of febrile rash cases reported | Number rubella IgM positive | Percentage IgM positive | Rubella Igm positive per 1 million population |
|---|---|---|---|---|
| 2007 | 99 | 20 | 20.2 | 6.8 |
| 2008 | 99 | 16 | 16.2 | 5.3 |
| 2009 | 157 | 7 | 4.5 | 2.3 |
| 2010 | 174 | 18 | 10.3 | 5.7 |
| 2011 | 228 | 20 | 8.8 | 6.3 |
| Total (2007–2011) | 757 | 81 | 10.7 | |

at these sites report febrile rash cases to the local government area disease surveillance and notification officers (DSNO). The local government areas DSNOs are expected to collect blood samples from the cases within 28 days of the onset of rash. One shortcoming of the surveillance system was that a lot of health facilities that were not part of the 130 focal sites do not routinely report the cases of febrile rash to the local government area disease surveillance and notification officers. The samples are sent to the measles national laboratory and tested for measles and rubella immunoglobulin G (IgM) antibodies using enzyme linked immunosorbent assay (ELISA). The Nigeria national measles and rubella laboratory operates according to World Health Organization standards and protocols. The samples are first tested for measles and rubella.

Recent infections of rubella are detected by positive IgM antibodies. However, there could be false positive rubella IgM results in patients with parvovirus B19, infectious mononucleosis or a positive rheumatoid factor (WHO, 2007). New cases of rubella per 1,000,000 population was calculated by dividing the number of reported rubella cases by the estimated population and multiplied by 1,000,000.

Data was analyzed using SAS 9.1.

## RESULTS

757 febrile rash cases were reported between 2007 and 2011. The mean age of the subjects was 7.5 years ($sd = 10.3$), while their age ranged from 0 to 80 years. Of the 757 febrile rash cases between 2007 and 2011, 81(10.7%) tested positive for rubella IgM. However, there were wide variations in the proportion that tested positive in different years, ranging from a high of 20.2% in 2007 to a low of 4.5% in 2009 (Table 1). The difference in the new rubella infection in the different years is statistically significant ($p = 0.0006$).

The result also showed that new rubella infection decreased from 6.81/1,000,000 population in 2007 to 2.28/1,000,000 in 2009 and increased to 6.34/1,000,000 in 2011 (Table 1).

Females (12.9%) were more likely to be infected with rubella compared to males (8.6%) with a relative risk of 1.5 (CI [0.98–2.28]). However, this higher risk is not statistically significant ($p = 0.057$) (Table 2).

Over the five years under review, there were more cases of rubella in people residing in rural areas (11.7%) compared to the urban areas (7.2%). The relative risk of having rubella

**Table 2  Demographics of rubella cases in Abia State, Nigeria, 2007–2011.**

|  | Total number of febrile rash cases reported | Number Rubella IgM positive | Percentage IgM positive | *P*-value | Relative risk (95% confidence interval) |
|---|---|---|---|---|---|
| **Sex** | | | | | |
| Female | 384 | 48 | 12.9% | 0.0571 | 1.50 (0.98–2.28) |
| Male | 373 | 33 | 8.6% | | |
| **Residence** | | | | | |
| Rural | 591 | 69 | 11.7% | 0.1015 | 1.62 (0.90–2.91) |
| Urban | 166 | 12 | 7.2% | | |
| **Age** | | | | | |
| <1 year | 117 | 3 | 2.6% | <0.0001 | 1.02[a] (0.56–1.86) |
| 1–4 years | 317 | 26 | 8.2% | | |
| 5–9 years | 150 | 28 | 18.7% | | |
| 10–14 years | 61 | 12 | 19.7% | | |
| ≥15 years | 101 | 11 | 10.9% | | |

**Notes.**
[a] Relative risk between two age groups (those less than 15 years and those 15 years and above).

is 1.6 times (CI [0.90–2.91]) as high in rural areas compared to urban areas, although this increased risk is not statistically significant ($p = 0.1$) (Table 2).

Furthermore, the proportion of people infected in the different age groups increased with increasing age until 14 years and decreased thereafter. Eighty six percent of all the rubella cases occurred in children between 0–14 years and 14% of the cases occurred in those 15 years and older. The difference in the new cases of rubella among the different groups is statistically significant ($p = <0.0001$) with most of the cases occurring between 5 and 14 years. However, when the people are divided into two age groups of less that 15 years and 15 years and greater, there is no statistical difference in the relative risk between the two age groups (RR = 1.02; 95% (CI [0.56–1.86])) (Table 2).

## DISCUSSION

Rubella vaccinations are not part of the routine vaccinations in Nigeria (*Onakewhor & Chiwuzie, 2011*). There has not been any attention on rubella as it is not considered a childhood killer disease. The major clinical concern for rubella is the risk of congenital rubella syndrome. Unfortunately, there is no data on the incidence of congenital rubella syndrome in Nigeria. Even the few studies on congenital rubella syndrome in Nigeria are only based on clinical diagnosis of rubella without any laboratory confirmation of such cases. A review of 16,394 births in southern Nigeria showed a congenital rubella syndrome rate of 1.16 per 1,000 live births (*George, Frank-Briggs & Oruamabo, 2009*). This is compared to an estimated rate of 0.8 per 1,000 live births in Kumasi Ghana (*Lawn et al., 2000*).

On average, 10.7% of the reported cases of febrile rash tested positive for rubella IgM. This is similar to 12% of reported cases of rash in Ethiopia and Akwa Ibom State in Nigeria that tested positive for rubella IgM respectively (*Mitiku et al., 2011*). The rate of rubella IgM positivity in children between 0–9 years in this study is 9.8% which is lower than the IgM positive rate of 45.2% among children between 0–10 years in Jos, North Central

Nigeria (*Junaid, Akpan & Olabode, 2011*). Furthermore, 3.9–5.2% of pregnant women in Nigeria were rubella IgM positive (*Onakewhor & Chiwuzie, 2011*; *Pennap et al., 2009*), while 3.4% of pregnant women in Western Sudan, were IgM positive (*Hamdan et al., 2011*). Apart from rubella, 10 percent (75) of the 757 reported cases of febrile rash was due to measles.

The result showed a low incidence of new rubella infection of 6.81/1,000,000 population in 2007 which decreased to 2.28/1,000,000 in 2009. However, it increased to 6.34/1,000,000 in 2011. The reason for the decrease in 2009 is not clear. There is no routine rubella vaccination in Nigeria and so this decrease could not have been due to increased rubella vaccination. This low incidence of rubella is incongruent with results of studies among women of child bearing age in Nigeria which showed that between 53% and 77% have had rubella (presence of rubella IgG antibodies in unimmunized women) (*Onakewhor & Chiwuzie, 2011*; *Bamgboye et al., 2004*; *Bukbuk, el Nafaty & Obed, 2002*; *Onyenekwe et al., 2000*). Thus the rubella surveillance system appears not to be robust enough to detect many cases of rubella. As many as 20–50% of rubella cases occur without a rash (*WHO, 2013a*; *WHO, 2011*). The proportion of new cases picked up by the surveillance system is still low compared to the prevalence of rubella IgG in the adult population. The assumption of a weak rubella surveillance system might also explain the reason for the huge difference in the rubella IgM positivity rate in children 0–9 years in this study (9.8%) and a population based study carried out in children 0–10 years in Jos, North Central Nigeria (45.2%) (*Junaid, Akpan & Olabode, 2011*).

Cumulatively, between 2007 and 2011, 41% of the new cases of rubella were male and 59% were female. Among IgM positive children in Jos, North Central Nigeria, 33% were male and 67% were females (*Junaid, Akpan & Olabode, 2011*). In Ethiopia, 54% of IgM positive cases were females (*Mitiku et al., 2011*).

Furthermore, the higher rate of rubella in rural areas compared to urban areas is consistent with findings in a similar study in southern Nigeria (*Enya et al., 2011*). This might be due to the fact that children who live in the rural areas are generally poorer, more malnourished and more susceptible to infections (*WHO/UNICEF, 2004*).

Eighty six percent of the cases occurred in children younger than 15 years old. This is comparable to 94.7% of IgM positive cases occurring in children younger than 15 years old in Ethiopia (*Mitiku et al., 2011*). This is also in agreement with earlier observations that rubella is a disease that mainly affects children (*WHO, 2011*). The low incidence of new cases of rubella after 15 years might be due to the fact that the susceptible population decreases with increasing age because a large proportion of the population would have been infected and developed immunity before adulthood. This is further buttressed by findings in Nigeria and Ouagadougou, Burkina Faso where rubella IgG antibodies in unvaccinated pregnant women and other women of child bearing age ranged from 53–77% in Nigeria while it is 77% in Ouagadougou, Burkina Faso (*Onakewhor & Chiwuzie, 2011*; *Bamgboye et al., 2004*; *Bukbuk, el Nafaty & Obed, 2002*; *Onyenekwe et al., 2000*; *Linguissi et al., 2012*). Bearing in mind that the rubella vaccination is not part of the routine vaccination in

Nigeria and Burkina Faso (*Abia State Government, 2013a*; *Tahita et al., 2013*), this means that these women had been infected at an earlier age and thereafter developed immunity.

## CONCLUSION

Rubella infection in Abia State, Nigeria is predominantly found in those who are younger than 15 years old. It is also more prevalent in females and in those living in rural areas of the state. Unfortunately, there is no surveillance of CRS in Nigeria and so the public health impact of rubella infection in the state is not known. Rubella surveillance should be enlarged to include surveillance for CRS. This can be done by making hospitals report suspected cases of CRS so that samples can be drawn from such babies for laboratory testing. Furthermore, it is obvious that rubella is prevalent in the state with associated risk of CRS. Efforts should be made to include rubella as a routine childhood vaccine in Nigeria. This should be followed by concomitant vaccination of non-pregnant women of child bearing age to avoid risk of CRS.

### Funding
The authors received no funding for this study.

### Competing Interests
The authors declare there are no competing interests.

### Author Contributions
- Chukwuemeka Anthony Umeh and Stella Chioma Onyi analyzed the data, wrote the paper, prepared figures and/or tables, reviewed drafts of the paper.

### Supplemental Information
Supplemental information for this article can be found online at http://dx.doi.org/10.7717/peerj.580#supplemental-information.

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
