# Peer review of "Case based rubella surveillance in Abia State, South East Nigeria, 2007–2011"

_PeerJ, doi:10.7717/peerj.580_

## Round 0.1 · original submission · Major Revisions

Please, pay particular attention to the reviewer 1's concerns

Reviewer 1 ·

Basic reporting

This study is an epidemiological report of rubella cases in the Abia State in Nigeria, as diagnosed by IgM testing on rash cases in the national laboratory. This follows the WHO guidelines for rubella diagnosis and reporting. The article is written clearly, but I have a number of minor comments:

1) The introduction may benefit by adding some context on vaccination and incidence of rubella in the rest of Nigeria.

2) CRS is mentioned throughout the paper and the authors should report what is the known or estimated incidence of CRS births in Nigeria, or whether CRS is systematically reported or not in Nigeria.

3) Line 52. I would like to ask the authors to substitute “burden” with a more neutral term (e.g. portion or percentage) to describe the young population.

Experimental design

This study is a simple epidemiological report of rubella cases diagnosed in Abia State, Nigeria.

4) Please add what kit was used for the ELISA test. This is a detail of interest for the international rubella community.

Validity of the findings

5) This is my most important comment. The interest of this study is diminished by the fact that we don’t know how many cases of rubella go undiagnosed. The authors say on line 99 that the real incidence could be more than twice than reported, and this is kind of a vague statement (3 times? 100 times?). Using the rate of rubella IgG seroprevalence in adult women reported in line 118 (53-77%) the authors should estimate the real incidence of rubella in Abia and give the readers a measure of the significance of the incidence reported in this study. Maybe the discussion then will be refocused on how many cases are missed, rather than how many are detected.

6) The interpretation of the results would benefit from some statistical analysis. For example, is the difference between males and females statistically significant? Is the incidence of rubella really different in the country compared to the cities?

7) In table 1, the authors noted a change in % of IgM+ rash cases. Can the authors comment on what was the diagnosis of the other cases? Were there measles outbreaks that would account for year-toyear variations?

8) Lines 94-95. Screening of pregnant women for rubella IgM is not recommended by WHO because of the possibility of false positives. Surely, this must be the case for a fraction of the IgM+ of the studies cited here. The authors should comment on this point.

Reviewer 2 ·

Basic reporting

The present manuscript entitled “The epidemiology of new rubella infections in Abia state, South East Nigeria” describes the prevalence of IgM antibodies by ELISA (therefore new infections) raised against Rubella in Abia State, Nigeria. The authors tested by c-ELISA 757 serum samples collected between 2007 and 2011 from human patient presenting febrile rash. They demonstrated that Rubella actively circulates in Abia state with new confirmed cases every year.

Experimental design

The authors tested by ELISA the presence of IgM antibodies in 757 serum samples.
In my opinion they should enlarge the study by including the neutralizing Abs (IgG) titres of women and perform by dedicate tools a proper risk analysis for fetal infection.
No statistical analysis has been performed. Is there any significant statistical difference of Rubella incidence within the years included in the survey? Similar results were also obtained in other surrounding countries? No cases of fetal malformation have been reported. Is that true?
As I said before, IgG titres by serum neutralization would donate a big value to the manuscript. The authors should reasonaby acknowledge that the effort spent in the study is somehow poor.

Validity of the findings

Findings are not surprising and the manuscript would benefit by inclyding the IgG titres.

Additional comments

The present manuscript entitled “The epidemiology of new rubella infections in Abia state, South East Nigeria” describes the prevalence of IgM antibodies by ELISA (therefore new infections) raised against Rubella in Abia State, Nigeria. The authors tested by c-ELISA 757 serum samples collected between 2007 and 2011 from human patient presenting febrile rash. They demonstrated that Rubella actively circulates in Abia state with new confirmed cases every year.In my opinion they should enlarge the study by including the neutralizing Abs (IgG) titres of women and perform by dedicate tools a proper risk analysis for fetal infection.
No statistical analysis has been performed. Is there any significant statistical difference of Rubella incidence within the years included in the survey? Similar results were also obtained in other surrounding countries? No cases of fetal malformation have been reported. Is that true?
As I said before, IgG titres by serum neutralization would donate a big value to the manuscript. The authors should reasonaby acknowledge that the effort spent in the study is somehow poor.Findings are not surprising and the manuscript would benefit by inclyding the IgG titres.

·

Basic reporting

Manuscript entitled " The epidemiology of new rubella infections in Abia state, South East Nigeriawhich you submitted to PPJ has been reviewed. In this manuscript, authors report their study of confirmed cases of rubella in Abia state Nigeria from 2007 to 2011 detected through Abia state Integrated Disease Surveillance and Response system. The study is interesting and could provide valuable information of the rubella virus infection . Further details of the followings should be provided and clarified:

1-The title should be more specific and well correlated with content.

2- Introduction: Line: 18: Rubella is an acute contagious viral infection that usually affects children and young adults. It is amild self-limited illness that presents with fever, maculopapular rash, malaise and mild conjunctivitis. Definition of rubella diseases do not include conjunctivitis. Please ad the phrase: Rubella virus (RV) is a highly infectious and teratogenic agent. Rubella virus infection during the first trimester of pregnancy may lead to fetal death or various birth defects, including deafness, cataracts, and heart disease, known as congenital rubella syndrome (CRS) [Hobman and Chantler,2007].

Experimental design

3- Material and Methods: Line 68: and rubella immunoglobulin G (IgM) antibodies using enzyme linked immunosorbent assay (ELISA). Wich? Siemens??? add to text.
4- The results was not structured properly. Table 1, 2 and 3 should be combined, which may improve clearness
5- Discussion: vaccination should be mentioned and the necessary measures for the control of rubella in the city.
5- The paper must be rewritten in grammatical English with the help of a native English speaking-scientist or scientific English editing service:

Validity of the findings

No comments

Additional comments

The paper must be approved after corrections.

---

## Round 0.2 · accepted · Accept

No more revisions are needed